# A Benchmark for Semi-Inductive Link Prediction in Knowledge Graphs

**Adrian Kochsiek**
University of Mannheim
Germany
akochsiek@uni-mannheim.de

**Rainer Gemulla**
University of Mannheim
Germany
rgemulla@uni-mannheim.de

## Abstract

Semi-inductive link prediction (LP) in knowledge graphs (KG) is the task of predicting facts for new, previously unseen entities based on context information. Although new entities can be integrated by retraining the model from scratch in principle, such an approach is infeasible for large-scale KGs, where retraining is expensive and new entities may arise frequently. In this paper, we propose and describe a large-scale benchmark to evaluate semi-inductive LP models. The benchmark is based on and extends Wikidata5M: It provides transductive, k-shot, and 0-shot LP tasks, each varying the available information from (i) only KG structure, to (ii) including textual mentions, and (iii) detailed descriptions of the entities. We report on a small study of recent approaches and found that semi-inductive LP performance is far from transductive performance on long-tail entities throughout all experiments. The benchmark provides a test bed for further research into integrating context and textual information in semi-inductive LP models.

## 1 Introduction

A knowledge graph (KG) is a collection of facts describing relations between real-world entities. Facts are represented in the form of subject-relation-object triples such as (*Dave Grohl*, *memberOf*, *Foo Fighters*). In this paper, we consider link prediction (LP) tasks, i.e., the problem of inferring missing facts in the KG. LP can be transductive (TD; all entities known a priori), semi-inductive (SI; some entities known a priori), and inductive (no entities known a priori). We concentrate on semi-inductive and transductive LP.

SI-LP focuses on modeling entities that are unknown or unseen during LP, such as out-of-KG entities (not part or not yet part of the KG) or newly created entities, e.g., a new user, product, or event. Such previously unknown entities can be handled by retraining in principle. For large-scale KGs, however, retraining is inherently expensive and new entities may arise frequently. Therefore, the goal of SI-LP is to avoid retraining and perform LP directly, i.e., to generalize beyond the entities seen during training.

To perform LP for unseen entities, context information about these entities is needed. The amount and form of context information varies widely and may take the form of facts and/or textual information, such as an entity mention and/or its description. For example, a new user in a social network may provide a name, basic facts such as gender or country of origin, and perhaps a textual self-description.

In this paper, we introduce the *Wikidata5M-SI* benchmark for SI-LP. Our benchmark is based on the popular Wikidata5M (Wang et al., 2021) benchmark and has four major design goals: (G1) It ensures that unseen entities are long tail entities since popular entities (such as, say, *Foo Fighters*) and/or types and taxons (such as human and organization) are unlikely to be unseen. (G2) It allows to evaluate each model with varying amounts of contextual facts (0-shot, few-shot, transductive), i.e., to explore individual models across a range of tasks. (G3) It provides a controlled amount of textual information (none, mention, full description), where each setting demands different modeling capabilities. Finally, (G4) the benchmark is large-scale so that retraining is not a suitable approach. All prior SI-LP benchmarks violate at least one of these criteria.

We report on a small experimental study with recent LP approaches. In general, we found that

1. SI performance was far behind TD performance in all experiments for long-tail entities,

2. there was generally a trade-off between TD and SI performance,

3. textual information was highly valuable,

4. proper integration of context and textual information needs further exploration, and

5. facts involving less common relations provided more useful context.

Our benchmark provides directions and a test bed for further research into SI-LP.

## 2   Related Work

Multiple SI-LP datasets have been proposed in the literature. The benchmarks of Daza et al. (2021), Albooyeh et al. (2020), and Galkin et al. (2021) are obtained by first merging the splits of smaller transductive LP datasets and subsequently sampling unseen entities uniformly to construct validation and test splits. These benchmarks do not satisfy goals G1–G4. Shi and Weninger (2018) follow a similar approach but focus on only 0-shot evaluation based on textual features. Xie et al. (2016) and Shah et al. (2019) select entities from Freebase with connection to entities in FB15k (Bordes et al., 2013), also focussing on 0-shot evaluation using rich textual descriptions. These approaches do not satisfy G2 and G3. Finally, Wang et al. (2019) and Hamaguchi et al. (2017) uniformly sample test triples and mark occurring entities as unseen. These approaches do not focus on long-tail entities (and, in fact, the accumulated context of unseen entities may be larger than the training graph itself) and they do not satisfy G1–G3.

There are also several of fully-inductive LP benchmarks (Teru et al., 2020; Wang et al., 2021) involving KGs. While SI-LP aims to connect unseen entities to an existing KG, fully-inductive LP reasons about a new KG with completely separate entities (but shared relations). We do not consider this task in this work.

## 3   The *Wikidata5M-SI* Benchmark

*Wikidata5M-SI* is based on the popular Wikidata5M (Wang et al., 2021) benchmark, which is induced by the 5M most common entities of Wikidata. Our benchmark contains transductive and semi-inductive valid/test splits; see Tab. 1 for an overview. Generally, we aimed to keep Wikidata5M-SI as close as possible to Wikidata5M. We did need to modify the original transductive valid and test splits, however, because they unintentionally contained both seen and unseen entities (i.e., these splits were not fully transductive). We

| | Train | Transductive | | Semi-inductive | |
| | | Valid | Test | Valid | Test |
|---|---|---|---|---|---|
| Triples | 20,600,187 | 4,983 | 4,977 | 5,500 | 5,500 |
| Entities | 4,593,103 | 7,768 | 7,760 | 3,722 | 3,793 |
| Entities unseen | - | 0 | 0 | 500 | 500 |
| Relations | 822 | 217 | 211 | 126 | 115 |

Table 1: Statistics of the Wikidata5M-SI splits.

did that by simply removing all triples involving unseen entities.

**Unseen entities.** To ensure that unseen entities in the semi-inductive splits are from the long tail (G1), we only considered entities of degree 20 or less. To be able to provide sufficient context for few-shot tasks (G2), we further did not consider entities of degree 10 or less. In more detail, we sampled 500 entities of degrees 11–20 (stratified sampling grouped by degree) for each semi-inductive split. All sampled entities, along with their facts, were removed from the train split. Note that these entities (naturally) have a different class distribution than all entities; see Sec. A.1 for details.

**Tasks and metrics.** For TD tasks, we follow the standard protocol of Wikidata5M. To construct SI tasks, we include 11 of the original facts of each unseen entity into its SI split; each split thus contains 5,500 triples. This enables up to 10-shot SI tasks (1 fact to test, up to 10 facts for context). For entities of degree larger than 11, we select the 11 facts with the most frequent relations; see Tab. 2 for an example. The rationale is that more common relations (such as *instanceOf* or *country*) may be considered more likely to be provided for unseen entities than rare ones (such as *militaryBranch* or *publisher*). We then construct a single $k$-shot task for each triple $(s, p, o)$ in the SI split as follows. When, say, $s$ is the unseen entity, we consider the LP task $(s, p, ?)$ and provide $k$ additional facts of form $(s, p', o')$ as context. Context facts are selected by frequency as above, but we also explored random and infrequent-relation context in our study. Models are asked to provide a ranking of predicted answers, and we determine the filtered mean reciprocal rank (MRR) and Hits@K of the correct answer $(o)$.

**Textual information.** For each entity, we provide its principal mention and a detailed description (both directly from Wikidata5M); see Tab. 2. This allows to differentiate model evaluation with varying amounts of textual information per entity (G3): (A) atomic, i.e., no textual information, (M) men-

| ID | Q18918 | | |
|---|---|---|---|
| Mention | Sam Witwer | | |
| Description | Samuel Stewart Witwer (born October 20, 1977) is an American actor and musician. He is known for portraying Crashdown in Battlestar Galactica, Davis Bloome in Smallville, Aidan Waite in Being Human, and Ben Lockwood in Supergirl. He voiced the protagonist Galen Marek / Starkiller in Star Wars: The Force Unleashed, the Son in Star Wars: The Clone Wars and Emperor Palpatine in Star Wars Rebels, both of which he has also voiced Darth Maul. | | |
| Context triples | instance of \| human | M: ∘ | D: ∘ |
| | country of citizenship \| United States of America | M: × | D: ∘ |
| | occupation \| musician | M: × | D: ✓ |
| | occupation \| actor | M: × | D: ✓ |
| | place of birth \| Glenview | M: × | D: × |
| | given name \| Samuel | M: ∘ | D: ✓ |
| | given name \| Sam | M: ✓ | D: ∘ |
| | cast member \| Battlestar Galactica | M: × | D: ✓ |
| | cast member \| Being Human - supernatural drama television series | M: × | D: ✓ |
| | cast member \| Star Wars: The Force Unleashed II | M: × | D: ∘ |
| | cast member \| The Mist | M: × | D: × |

Table 2: Example of an entity from the semi-inductive validation set of Wikidata5M-SI. For each triple, we annotated whether the answer is contained in (✓), deducible from (∘), or not contained in (×) mention (M) or description (D).

tions only, and (D) detailed textual descriptions as in (Kochsiek et al., 2023). This differentiation is especially important in the SI setting, as detailed text descriptions might not be provided for unseen entities and each setting demands different modeling capabilities. In fact, (A) performs reasoning only using graph structure, whereas (D) also benefits from information extraction to some extent. We discuss this further in Sec. 5.

## 4 Semi-Inductive Link Prediction Models

We briefly summarize recent models for SI-LP; we considered these models in our experimental study.

**Graph-only models.** *ComplEx* (Trouillon et al., 2016) is the best-performing transductive KGE model on Wikidata5M (Kochsiek et al., 2022). To use ComplEx for SI-LP, we follow an approach explored by Jambor et al. (2021). In particular, we represent each entity as the sum of a local embedding (one per entity) and a global bias embedding. For 0-shot, we solely use the global bias for the unseen entity. For k-shot, we obtain the local embedding for the unseen entity by performing a single training step on the context triples (keeping all other embeddings fixed). An alternative

approach is taken by *oDistMult-ERAvg* (Albooyeh et al., 2020), which represents unseen entities by aggregating the embeddings of the relations and entities in the context.[1] A more direct approach is taken by *HittER* (Chen et al., 2021), which contextualizes the query entity with its neighborhood for TD-LP. The approach can be used for SI-LP directly by using a masking token (akin to the global bias above) for an unseen entity. We originally planned to consider *NodePiece* (Galkin et al., 2021) (entity represented by a combination of anchor embeddings) and *NBFNet* (Zhu et al., 2021) (a GNN-based LP model); both support SI-LP directly. However, the available implementations did not scale to Wikidata5M-SI (out of memory).[2]

**Text-based models.** As a baseline approach to integrate textual information directly into KGE models, we consider the approach explored in the

---

[1]To address the high memory footprint (Galkin et al., 2021) of oDistMult-ERAvg, we extend it with neighborhood sampling.

[2]For NBFNet (Zhu et al., 2021), the large memory footprint is inherent to the model; it is a full-graph GNN and hard to scale. For NodePiece (Galkin et al., 2021), however, the problem mainly lies in the expensive evaluation. All intermediate representations are precomputed, leading to a large memory overhead.

WikiKG90M benchmark (Hu et al., 2021); see Sec. A.2 for details. The remaining approaches are purely textual. *SimKGC* (Wang et al., 2022) utilizes two pretrained BERT Transformers: one to embed query entities (and relations) based on their mention or description, and one for tail entities. Using a contrastive learning approach, it measures cosine similarity between both representations for ranking. *KGT5* (Saxena et al., 2022) is a sequence-to-sequence link prediction approach, which is trained to generate the mention of the answer entity using the mention or description of the query entity and relation as input. Both approaches support 0-shot SI-LP when textual information is provided for the query entity. They do not utilize additional context, however, i.e., do not support k-shot SI-LP. *KGT5-context* (Kochsiek et al., 2023) is an extension of KGT5, which extends the input of KGT5 by the one-hop neighborhood of the query entity and consequently supports k-shot LP directly.

# 5  Experimental Study

We evaluated all presented baseline models in the TD and SI setting on the atomic, mentions, and descriptions dataset. Further, we evaluated in detail which context was most useful and what information was conveyed by textual mentions and descriptions.

**Setup.** Source code, configuration, and the benchmark itself are available at `https://github.com/uma-pi1/wikidata5m-si`. For further details on hyperparameter tuning and training see Sec. A.3.

**Main results.** Transductive and SI performance in terms of MRR of all models is presented in Tab. 3; Hits@K in Tab. 7-9 (Sec. A). Note that overall transductive performance was oftentimes below best reported SI performance. This is due to varying degrees of query entities between both settings. Typically, models perform better predicting new relations for an entity (e.g., the birthplace) than predicting additional objects for a known relation (e.g., additional awards won by a person) (Saxena et al., 2022; Kochsiek et al., 2023). For a direct comparison between both settings, we additionally report TD performance on long tail query entities.[3]

**Atomic.** TD performance on the long tail was considerably higher than SI performance. As no in-

formation was provided for unseen entities, 0-shot was not reasonably possible. Without text-based information, context was a necessity. A simple neighborhood aggregation—entity-relation average (ERAvg)—offered the best integration of context.

**Mentions.** Integrating mentions did not improve performance on its own, as provided text information was still limited. However, additionally providing context information during inference (KGT5-context) simplified the learning problem and improved TD performance significantly. But for 0-shot, the limited text information provided with mentions allowed for reasonable performance. To analyze what information is conveyed for 0-shot, we annotated 100 valid triples; see Tab. 4. In 10% of cases, the answer was already contained in the mention, and it was deducible in at least 7%. This enabled basic reasoning without any further information. In contrast to the TD setting, KGT5 outperformed its context extension. KGT5-context was reliant on context which was lacking especially during 0-shot. This showed a trade-off between best performance in the SI and TD setting. This trade-off could be mitigated by applying (full and partial) context hiding. With such adapted training, KGT5-context reached a middle ground with a transductive MRR of 0.366 and 0-shot MRR of 0.283.[4] However, even with full context (10-shot), performance was still only on par with KGT5. Therefore, context information did not bring any further benefits when text was provided.

**Descriptions.** Further, integrating descriptions improved performance for both settings, TD and SI, considerably; see Tab. 3. Similar to the mentions-only setting, KGT5-context performed best in TD and KGT5 in the SI setting. Applying the same trade-off with context-hiding reached a middle ground with 0.418 TD-MRR and 0.449 SI-MRR.

Descriptions were very detailed and partially contained the correct answer as well as the same information as contained in context triples; see Tab. 4. Therefore, performance did not further improve with context size. In such cases, models mainly benefit from information extraction capabilities. To judge how much information extraction helps, we grouped performance of KGT5+description in the 0-shot setting on validation data into the groups *contained*, *deducible* and *not contained* in descrip-

---

[3]We define long tail query entities as entities with degree ≤ 10 as in the SI setting.

[4]In 25%/25%/50% of cases, we hid the full context/sampled between 1-10 neighbors/used the full context, respectively.

| Model | Transductive | | Semi-inductive (num. shots) | | | | | Pre-trained |
|---|---|---|---|---|---|---|---|---|
| | All | Long tail | 0 | 1 | 3 | 5 | 10 | |
| ComplEx + Bias + Fold in (Jambor et al., 2021) | 0.308 | 0.523 | 0.124 | 0.151 | 0.176 | 0.190 | 0.206 | no |
| DistMult + ERAvg (Albooyeh et al., 2020) | 0.294 | 0.512 | - | 0.171 | 0.246 | 0.295 | 0.333 | no |
| HittER (Chen et al., 2021) | 0.284 | 0.512 | 0.019 | 0.105 | 0.153 | 0.179 | 0.221 | no |
| DistMult + ERAvg + Mentions | 0.299 | 0.535 | - | 0.187 | 0.235 | 0.258 | 0.280 | yes |
| SimKGC (mentions only) | 0.212 | 0.361 | 0.220 | - | - | - | - | yes |
| KGT5 (Saxena et al., 2022) | 0.281 | 0.542 | 0.310 | - | - | - | - | no |
| KGT5-context (Kochsiek et al., 2023) | 0.374 | 0.678 | 0.220 | 0.217 | 0.236 | 0.259 | 0.311 | no |
| DistMult + ERAvg + Descriptions | 0.313 | 0.585 | - | 0.278 | 0.281 | 0.285 | 0.292 | yes |
| SimKGC + Descriptions (Wang et al., 2022) | 0.353 | 0.663 | 0.403 | - | - | - | - | yes |
| KGT5 + Descriptions (Kochsiek et al., 2023) | 0.364 | 0.728 | **0.470** | - | - | - | - | no |
| KGT5-context + Descriptions (Kochsiek et al., 2023) | **0.420** | 0.777 | 0.417 | 0.420 | 0.416 | 0.420 | 0.437 | no |

Table 3: Transductive and semi-inductive link prediction results in terms of MRR on the dataset Wikidata5M-SI. The first group presets results on the atomic, the second on the mentions and the third on the descriptions dataset. Best per TD/SI in bold. Best per group underlined.

| | Mention | Description |
|---|---|---|
| Contained | 10% | 44% |
| Deducible | 7% | 10% |
| Not contained | 83% | 46% |

Table 4: Information about a query answer contained in mentions and descriptions. Annotated for 100 sampled triples from 0-shot valid. For an example, see Tab. 2.

| Context selection | 1 | 3 | 5 |
|---|---|---|---|
| Most common | 0.217 | 0.236 | 0.259 |
| Least common | 0.253 | 0.273 | 0.290 |
| Random | 0.237 | 0.260 | 0.281 |

Table 5: Influence of context selection. Semi-inductive test MRR of KGT5-context.

tion; see Fig. 1 in Sec. A. When contained, the correct answer was extracted in $\approx 70\%$ of cases.

**Context selection.** We selected the most common relations as context triples so far, as this may be a more realistic setting. To investigate the effect of this selection approach, we compared the default selection of choosing *most common* relations to *least common* and *random*. Results for KGT5-context are shown in Tab. 5; for all other models in Tab. 10 in Sec. A. We found that the less common the relations of the provided context, the better the SI performance. More common context relations often described high-level concepts, while less common provided further detail; see the example in Tab. 2. While more common context may be more readily available, less common context was more helpful to describe a new entity.

## 6  Conclusion

We proposed the new WikiData5M-SI large-scale benchmark for semi-supervised link prediction. The benchmark focuses on unseen entities from the long tail and allows to evaluate models with varying and controlled amounts of factual and textual context information. In our experimental evaluation, we found that semi-inductive LP performance fell behind transductive performance for long-tail entities in general, and that detailed textual information was often more valuable than factual context information. Moreover, current models did not integrate these two types of information adequately, suggesting a direction for future research.

### Limitations

This study was performed on Wikidata5M-SI, i.e., a subset of a single knowledge graph. Model performance and insights may vary if graph structure and/or availability and usefulness of mentions and description is different. In particular, the entity descriptions provided with Wikidata5M-SI partly contained information relevant for link prediction so that models benefited from information extraction capabilities.

### Ethics Statement

This research adapts publicly available data, benchmarks, and codebases for evaluation. We believe that this research was conducted in an ethical manner in compliance with all relevant laws and regulations.

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

## A Appendix

### A.1 Distribution of Unseen Entities

Long-tail entities have a different distribution than entities from the whole KG; see Tab. 6 for an overview of the distribution shift for the top 10 entity types. This difference is natural. In particular, high-degree entities in a KG such as Wikidata often refer to types/taxons (e.g, human, organization, ...) as well as popular named entities (e.g., Albert Einstein, Germany, ...). These entities are fundamental to the KG and/or of high interest and have many facts associated with them. For this reason, they do not form suitable candidates for benchmarking unseen or new entities. In addition, removing high-degree entities for the purpose of evaluating SI-LP is likely to distort the KG (e.g., consider removing type "human" or "Germany"). In contrast, Wikidata5M-SI focuses on entities for which knowledge is not yet abundant: long-tail entities are accompanied by no or few facts (at least initially) and our SI-LP benchmark tests reasoning capabilities with this limited information.

### A.2 Integrating Text into KGE Models

To integrate text into traditional KGE models, we follow the baseline models of the WikiKG90M link prediction challenge (Hu et al., 2021). We embed mentions combined with descriptions using MPNet (Song et al., 2020), concatenate the resulting descriptions embedding with the entity embedding, and project it with a linear layer for the final representation of the entity. In combination with oDistMult-ERAvg (Albooyeh et al., 2020), we apply the aggregation of neighboring entities and relations on the entity embedding part only. The resulting aggregation is then concatenated with its description and finally projected.

This approach is closely related to BLP (Daza et al., 2021). The main differences to BLP are:

1. Hu et al. (2021) use MPNet, BLP uses BERT.

2. In combination with DistMult-ERAvg, we concatenate a learnable "structural embedding" to the CLS embedding of the language model, whereas BLP does not.

### A.3 Experimental Setup

For hyperparameter optimization for ComplEx (Trouillon et al., 2016), DistMult (Yang et al., 2015), and HittER (Chen et al., 2021), we used the multi-fidelity approach GraSH (Kochsiek et al., 2022) implemented in LibKGE (Broscheit et al., 2020) with 64 initial trials and trained for up to 64 epochs. For fold-in, we reused training hyperparameters and trained for a single epoch on the provided context. For text-based approaches, we used the hyperparameters and architectures proposed by the authors for the transductive split of Wikidata5M. We trained on up to 5 A6000-GPUs with 49GB of VRAM.

| WikidataID | Mention | All entities | Long-tail entities |
|---|---|---|---|
| Q5 | human | 39% | 61% |
| Q11424 | film | 3% | 8% |
| Q484170 | commune of France | 1% | 7% |
| Q482994 | album | 3% | 1% |
| Q16521 | taxon | 9% | 1% |
| Q134556 | single | 1% | 1% |
| Q747074 | commune of Italy | 0% | 1% |
| Q2074737 | municipality of Spain | 0% | 1% |
| Q571 | book | 1% | 1% |
| Q7889 | video game | 1% | 1% |

Table 6: Distribution of top 10 entity types over long-tail entities with degree between 11 and 20 compared to all entities.

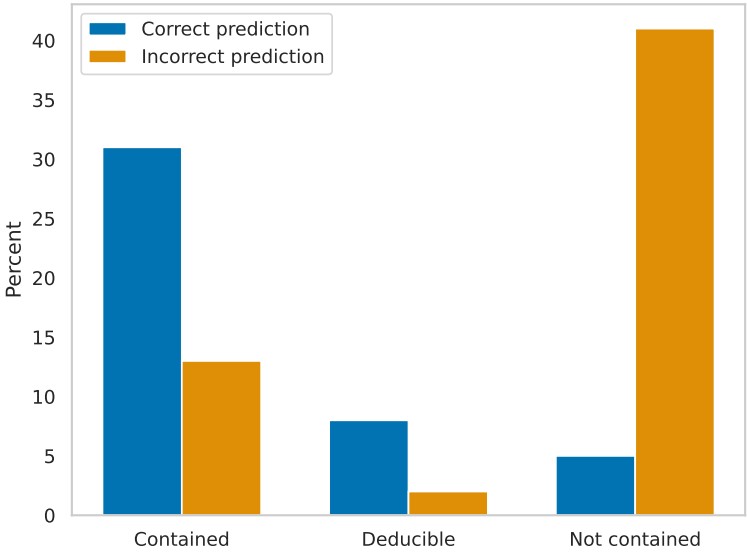

Figure 1: Number of correct (rank=1) and incorrect predictions by KGT5+descriptions on annotated examples per annotation label.

| | | **Semi-inductive (num. shots)** | | | | |
|---|---|---|---|---|---|---|
| **Model** | **Trans.** | **0** | **1** | **3** | **5** | **10** |
| Complex + Bias + Fold in (Jambor et al., 2021) | 0.260 | 0.058 | 0.097 | 0.118 | 0.124 | 0.132 |
| DistMult + ERAvg (Albooyeh et al., 2020) | 0.237 | - | 0.115 | 0.151 | 0.185 | 0.209 |
| HittER (Chen et al., 2021) | 0.234 | 0.005 | 0.076 | 0.115 | 0.132 | 0.153 |
| DistMult + ERAvg + Mentions | 0.239 | - | 0.106 | 0.142 | 0.153 | 0.167 |
| SimKGC (mentions only) | 0.182 | 0.187 | - | - | - | - |
| KGT5 (Saxena et al., 2022) | 0.249 | 0.263 | - | - | - | - |
| KGT5-context (Kochsiek et al., 2023) | 0.347 | 0.184 | 0.177 | 0.195 | 0.218 | 0.263 |
| DistMult + ERAvg + Descriptions | 0.252 | - | 0.152 | 0.153 | 0.153 | 0.161 |
| SimKGC + Descriptions (Wang et al., 2022) | 0.311 | 0.349 | - | - | - | - |
| KGT5 + Descriptions | 0.332 | 0.430 | - | - | - | - |
| KGT5-context + Descriptions | 0.400 | 0.379 | 0.382 | 0.373 | 0.378 | 0.393 |

Table 7: Transductive and semi-inductive link prediction results in terms of H@1 on the dataset Wikidata5M-SI.

| Model | Trans. | Semi-inductive (num. shots) | | | | |
|---|---|---|---|---|---|---|
| | | **0** | **1** | **3** | **5** | **10** |
| ComplEx + Bias + Fold in (Jambor et al., 2021) | 0.337 | 0.165 | 0.180 | 0.202 | 0.219 | 0.242 |
| DistMult + ERAvg (Albooyeh et al., 2020) | 0.328 | - | 0.190 | 0.292 | 0.352 | 0.401 |
| HittER (Chen et al., 2021) | 0.309 | 0.013 | 0.109 | 0.158 | 0.188 | 0.242 |
| DistMult + ERAvg + Mentions | 0.332 | - | 0.239 | 0.289 | 0.314 | 0.340 |
| SimKGC (mentions only) | 0.223 | 0.227 | - | - | - | - |
| KGT5 (Saxena et al., 2022) | 0.296 | 0.332 | - | - | - | - |
| KGT5-context (Kochsiek et al., 2023) | 0.390 | 0.236 | 0.234 | 0.257 | 0.278 | 0.335 |
| DistMult + ERAvg + Descriptions | 0.344 | - | 0.368 | 0.373 | 0.378 | 0.380 |
| SimKGC (Wang et al., 2022) | 0.367 | 0.421 | - | - | - | - |
| KGT5 + Descriptions | 0.385 | 0.490 | - | - | - | - |
| KGT5-context + Descriptions | 0.432 | 0.441 | 0.443 | 0.443 | 0.447 | 0.463 |

Table 8: Transductive and semi-inductive link prediction results in terms of H@3 on the dataset Wikidata5M-SI.

| Model | Trans. | Semi-inductive (num. shots) | | | | |
|---|---|---|---|---|---|---|
| | | **0** | **1** | **3** | **5** | **10** |
| ComplEx + Bias + Fold in (Jambor et al., 2021) | 0.387 | 0.231 | 0.245 | 0.282 | 0.309 | 0.336 |
| DistMult + ERAvg (Albooyeh et al., 2020) | 0.389 | - | 0.270 | 0.409 | 0.493 | 0.564 |
| HittER (Chen et al., 2021) | 0.376 | 0.050 | 0.157 | 0.226 | 0.270 | 0.359 |
| DistMult + ERAvg + Mentions | 0.411 | - | 0.320 | 0.392 | 0.440 | 0.478 |
| SimKGC (mentions only) | 0.266 | 0.283 | - | - | - | - |
| KGT5 (Saxena et al., 2022) | 0.344 | 0.398 | - | - | - | - |
| KGT5-context (Kochsiek et al., 2023) | 0.423 | 0.293 | 0.295 | 0.310 | 0.336 | 0.400 |
| DistMult + ERAvg + Descriptions | 0.425 | - | 0.465 | 0.472 | 0.484 | 0.491 |
| SimKGC (Wang et al., 2022) | 0.432 | 0.504 | - | - | - | - |
| KGT5 + Descriptions | 0.416 | 0.544 | - | - | - | - |
| KGT5-context + Descriptions | 0.455 | 0.484 | 0.489 | 0.489 | 0.495 | 0.516 |

Table 9: Transductive and semi-inductive link prediction results in terms of H@10 on the dataset Wikidata5M-SI.

| Model | Context selection | 1 | 3 | 5 |
|---|---|---|---|---|
| ComplEx + fold-in | Most common | 0.151 | 0.161 | 0.168 |
| | Least common | **0.166** | 0.185 | 0.195 |
| | Random | 0.164 | **0.187** | **0.196** |
| DistMult + ERAvg | Most common | 0.171 | 0.246 | 0.295 |
| | Least common | **0.217** | 0.299 | **0.323** |
| | Random | 0.215 | **0.303** | 0.318 |
| oDistMult + ERAvg + Mentions | Most common | 0.187 | 0.235 | 0.258 |
| | Least common | **0.237** | **0.274** | **0.279** |
| | Random | 0.232 | 0.265 | 0.272 |
| HittER | Most common | 0.105 | 0.153 | 0.179 |
| | Least common | **0.151** | **0.195** | **0.216** |
| | Random | 0.136 | 0.190 | 0.206 |
| KGT5-context | Most common | 0.217 | 0.236 | 0.259 |
| | Least common | **0.253** | **0.273** | **0.290** |
| | Random | 0.237 | 0.260 | 0.281 |
| KGT5-context + Desc. | Most common | 0.420 | 0.416 | 0.420 |
| | Least common | **0.423** | 0.424 | **0.430** |
| | Random | 0.422 | **0.430** | **0.430** |

Table 10: Influence of context selection. Semi-inductive test MRR. Best per model in bold.