# OpenReview forum: "A Benchmark for Semi-Inductive Link Prediction in Knowledge Graphs"
_EMNLP/2023/Conference — EMNLP 2023 Findings_

### Official Review · Reviewer_onsE · 2023-07-31

**Soundness:** 4

**Excitement:**

4: Strong: This paper deepens the understanding of some phenomenon or lowers the barriers to an existing research direction.

**Missing References:**

You might want to add https://arxiv.org/pdf/2301.00716.pdf as a complementary approach: they study scarcity, noisy text, 0-shot only SI-LP, also using a biased sampling approach for the training/test split.


**Paper Topic And Main Contributions:**

The paper proposes a new benchmark to study transductive and semi-inductive link prediction models. In a semi-inductive setting, links to a known graph for new unseen entities must be predicted. The dataset is based on Wikidata5m and offers a transductive (known entities), 0-shot (only textual features), and k-shot (text + k 1-hop neighbourhood triples) settings. The text is either only a mention (single phrase, usually the name of the entity) or a verbose description. The training/test-split is created by sampling from all 11-20-degree entities and setting them aside. Model performance is measured using MRR/Hits@k. The authors test multiple models to set up the SOTA baseline performance on the new benchmark. The main findings are: transductive is easier than semi-inductive, a trade-off exists between transductive and semi-inductive performance, high-quality text directly influences model performance, more study of hybrid models is required, spurious relations offer better information than general ones.

**Questions For The Authors:**

A. Table 2: Do you have any intuition/explanation as to why KGT5 + Descriptions works better (by far) than KGT5 + context + Descriptions? Does the encoder struggle with long/complex inputs? Is there any ablation how context length (descriptions + verbalized context) affects performance?
B. I have not understood "dropout" and the "trade-off" described (e.g. in 266). Dropout as a regularization technique? Can you elaborate on that?
C. Do you have any insights into whether any measurable biases exist when sampling only from the 11-20 degree entities? Are some entity types over- or under-represented? Is this in any way problematic?
D. You mentioned that most of the required information is already present in the descriptions and IR approaches are required for accessing that. Do you have any concrete ideas about how to tackle this? If it is “easy” to do - should there be a baseline in your paper showing how hard accessing this information is?
E. Talking about "realistic" for the selection of the k-shot triples: I think this is a problematic word in general because what "realistic" constitutes always depends on the context (for example, you assume the availability of high quality text to be "realistic"). Maybe it is simply the "hardest" setting (as shown in Table 3)?

**Reasons To Accept:**

The proposed benchmark fills a gap which is not suitably filled by current benchmarks, which study semi-inductive link prediction with text. The closest semi-inductive benchmark which offers multiple text modes by Daza et al. is neither as large-scale nor offers a few-shot evaluation. The study of hybrid models which combine text (calling for NLP) and knowledge graphs is a prevailing topic and fits with the EMNLP. The paper is well written and easy to follow.

**Reasons To Reject:**

The models provide an insight into the expected SOTA performance; however, there could have been more. It is understandable that computational constraints may prevent some models from working out-of-the-box, but it was not made clear if these are inherent problems concerning the run-time (algorithmically) or just poor implementations. BLP, for example, should work - at least offline.


**Reproducibility:**

4: Could mostly reproduce the results, but there may be some variation because of sample variance or minor variations in their interpretation of the protocol or method.

**Reviewer Confidence:**

4: Quite sure. I tried to check the important points carefully. It's unlikely, though conceivable, that I missed something that should affect my ratings.

**Typos Grammar Style And Presentation Improvements:**

Other remarks are marked by the line number or table/figure reference:

29: Are they really the most relevant? Transductive is definitely the most prevalent. Such statements, without any justification at least, are always risky.

43: Long-tail: its definition first appears on pg. 3 - maybe add a (sparsely connected) to clarify.
Up to 156: - A clarification about what "mention" means would have been nice for people unfamiliar with the internal naming of Wikidata properties (a simple example completely suffices). Mentions (e.g. in entity-linking) are usually multiple phrases associated to a single entity.

163: All employed models are "embedding" based - maybe better "structural", "graph-based", or "graph-only"?

189: (As previously mentioned) Are only their implementations running out of memory or is this a theoretical constraint inherent to their methods? If it is not a theoretical constraint, then it is a bit strange to say "We would have, but it would have caused additional work". Apart from that: wouldn't it be a nice way to underline the computational challenge of (your) large-scale dataset(s)

195: I don't like the explanation: It is using two BERTs, for contrastive learning: one encodes "h [SEP ] r" and one encodes tail entities; it measures cosine similarity (the dot-product calculation is a computational trick)

221: Wording unclear: Any textual information? I would guess that verbose, curated, high quality text is highly valuable in contrast to the simple mention phrase?

231: This contradicts finding (1)?

Table 4: Small typo: "inductve"

313: Isn't this just saying: add more data to be better (which is never surprising for deep learning models)? My point being: to reach transductive setting performance, we only need to add quasi-transductive data (triples) to the data instead of working on the models? This falls a bit short, IMHO.

---

> ### Author Rebuttal · Authors · 2023-08-28
>
> Thank you for your valuable comments and constructive criticism. We provide thoughts on each point below, and we will update the paper accordingly.
>
> > I have not understood "dropout" and the "trade-off" described (e.g. in 266). Dropout as a regularization technique? Can you elaborate on that?
>
> Thank you for pointing this out. Our description is indeed misleading, a better term may be "neighborhood hiding" (we'll change that in the paper).
>
> When "neighborhood hiding" is enabled, we (i) remove the complete neighborhood from the prompt (w.p. 25%), (ii) sample the neighborhood down to 1-10 triples (25\%), or (iii) leave the neighborhood untouched (50%). The idea is to provide the model with less neighborhood information during training by modifying the prompt. This makes the prompts closer to semi-inductive prompts, but takes them further away from transductive prompts.
>
> The described tradeoff is between the increased semi-inductive performance of KGT5-context and the (slightly) reduced transductive performance when using neighborhood hiding.
>
> > Table 2: Do you have any intuition/explanation as to why KGT5 + Descriptions works better (by far) than KGT5 + context + Descriptions?
>
> KGT5-context seems to be heavily reliant on the provided neighborhood (up to 100), i.e., its focus on neighborhood seems to be larger than the focus on the description. Since neighborhood is missing or reduced (up to 10) in k-shot LP, performance drops considerably. The neighborhood hiding technique described above is a simple baseline approach that aims to mitigate this effect, but it's clearly not particularly effective and should be explored further in future work.
>
> >  Is there any ablation how context length (descriptions + verbalized context) affects performance?
>
> In the context of few-shot prediction, both full neighborhood and full description typically fit into the input length. For this reason, "k" can be seen as a measure of context length. We did not perform any further ablation.
>
> The authors of KGT5-context provided anecdotal results with larger context sizes, however, reporting better performance in general. In our study, we used the default settings of KGT5-context.
>
> > Do you have any insights into whether any measurable biases exist when sampling only from the 11-20 degree entities? Are some entity types over- or under-represented?
>
> Yes, they are; this bias is by design. In particular, high-degree entities in a KG such as Wikidata often refer to types/taxons (e.g, human, organization, ...) as well as popular named entities (e.g., Albert Einstein, Germany, ...). These entities are fundamental to the KG and/or of high interest and have many facts associated with them. For this reason, they do not form suitable candidates for unseen or new entities. In addition, removing high-degree entities for the purpose of evaluating SI-LP is likely to distort the KG (e.g., consider removing type "human" or "Germany"). In contrast, our benchmark focuses on entities for which knowledge is not yet abundant: long-tail entities are accompanied by no or few facts (at least initially) and our SI-LP benchmark tests reasoning capabilities with this limited information.
>
> The table below shows the fraction of entities of a certain type across all entities and across entities with degree 11-20; the shift in distribution is apparent (e.g., more humans, less taxons). If deemed useful, we will add the table to the appendix of the paper.
>
> |   WikidataID  | Mention   | Pct full graph    | Pct degree 11-20  |
> |   ------  |   ------  |   -------:    |   ------: |
> | Q5 | human | 36%   | 61% |
> | Q16521 | taxon | 9%   | 1% |
> | Q13406463 | Wikimedia list article | 3%   | 0% |
> | Q11424 | film | 3%   | 8% |
> | Q486972 | human settlement | 3%   | 0% |
>
>
> > If it \[accessing information from descriptions\] is “easy” to do - should there be a baseline in your paper showing how hard accessing this information is?
>
> KGT5 + description in the zero-shot setting essentially serves as an LLM baseline. The paper currently does not spell this out / provide more insight.
>
> For the annotated examples presented in Tab. 5, we grouped the zero-shot performance by information in the description and obtained:
>
> | Setting       | Answer on rank 1 |
> |---------------|   ------: |
> | contained     |   75% |
> | deducible     |   50% |
> | not contained |   0%  |
>
> We will extend this analysis to all annotated triples mentioned Tab. 3 and include it in the paper.
>
> >  Selection of k-shot triples: "realistic" constitutes always depends on the context. Maybe it is simply the "hardest" setting?
>
> Yes, good point!
>
> > Computational constraints may prevent some models from working out-of-the-box, but it was not made clear if these are inherent problems concerning the run-time (algorithmically).
>
> For NBFNet \[1\] the large memory footprint is inherent to the model; it is a full-graph GNN and hard to scale. For NodePiece \[2\], however, the problem mainly lies in the expensive evaluation code that is available; it is more likely to be addressable. We will clarify this in the paper.
>
> > BLP, for example, should work - at least offline.
>
> BLP is closely related to the DistMult+Description approach presented in the paper; we chose the latter as it is also used as a baseline in \[3\]. The main differences to BLP are:
>
> (i) \[3\] uses MPNet, BLP uses BERT.
>
> (ii) \[3\] concatenates a learnable "structural embedding" to the CLS embedding of the language model, whereas BLP does not.
>
> We will clarify this in the paper.
>
> > 163: All employed models are "embedding" based - maybe better "structural", "graph-based", or "graph-only"?
>
> Indeed, thank you!
>
> > 231: This contradicts finding (1)?
>
> Thank you, again! Our presentation is misleading indeed. Finding (1) shoud read "semi-inductive performance is far behind transductive performance **for long-tail entities**."
>
> Transductive evaluation is performed on query entities with varying degree, while SI evaluation is only performed on low-degree query entities. The finding is supposed to say that transductive performance on low-degree query entities is considerably higher than SI performance on low-degree entties (see column long-tail in Tab. 2).
>
> > You might want to add https://arxiv.org/pdf/2301.00716.pdf
>
> Thank you!
>
> \[1\] Zhu, Zhaocheng, et al. "Neural bellman-ford networks: A general graph neural network framework for link prediction." Advances in Neural Information Processing Systems 34 (2021): 29476-29490
>
> \[2\] Galkin, Mikhail, et al. "NodePiece: Compositional and Parameter-Efficient Representations of Large Knowledge Graphs." International Conference on Learning Representations. 2021.
>
> \[3\] Hu, Weihua, et al. "OGB-LSC: A Large-Scale Challenge for Machine Learning on Graphs." Thirty-fifth Conference on Neural Information Processing Systems Datasets and Benchmarks Track (Round 2). 2021.

---

### Official Review · Reviewer_jB8R · 2023-08-05

**Soundness:** 3

**Excitement:**

3: Ambivalent: It has merits (e.g., it reports state-of-the-art results, the idea is nice), but there are key weaknesses (e.g., it describes incremental work), and it can significantly benefit from another round of revision. However, I won't object to accepting it if my co-reviewers champion it.

**Paper Topic And Main Contributions:**

Summary:

The paper introduces a benchmark for semi-inductive link prediction (SI-LP) in knowledge graphs (KG). The proposed dataset is based on an existing (transductive and inductive) link prediction dataset, Wikidata5M. Unlike the inductive setting where all the test entities are unseen during training, the semi-inductive setting addresses the case where test entities contain a mix of seen and unseen entities, e.g., new entities joining the KG. The benchmark also enables evaluating models on zero-shot and k-shot link prediction (0<=k<=10).

The paper also evaluates existing models (possibly adapted for SI-LP) on the benchmark and presents their findings, such as the importance of textual information for the task, compared with the context information.

Contributions:
1. A benchmark for semi-inductive link prediction
2. A detailed comparison of existing models on the benchmark
3. A comparison of the impact of textual description vs context information on the SI-LP task
4. A finding that less common relations are more informative as context information

Strength:
1. The paper makes interesting observations about the importance of various sources of information (i.e., context, mention, description) for the link prediction task.
2. The experiments provide sufficient justifications for the observations made in the paper.

Weakness:
1. The paper does not provide enough motivation for why the SI-LP task is important, and why we need a new benchmark for it when transductive and inductive benchmarks already exist in the literature. Also, the description of semi-inductive is a bit ambiguous. It is described with "some entities known apriori", but it is not very clear whether each test triple has a k-shot (possibly zero) entity and a known entity (frequency > 20). Also, it is unclear what advantage we get from analyzing few-shot (k>0) link prediction compared to the transductive setting.


**Questions For The Authors:**

1. Does semi-inductive mean exactly one entity (either head or tail) is unseen during training in the zero-shot setting? Similarly, in few-shot, does exactly one entity is a low-frequency entity?
2. What is the average degree or degree distribution of the entities in the dataset?
3. Removing triples from a split can lead to a cascading effect, i.e., a reduction in other entities' frequencies. Was it one-shot removal or any measures were taken to handle the cascading effect?


**Reasons To Accept:**

1. The paper proposes a new benchmark for zero-shot and few-shot link prediction tasks that can promote further research in this direction.
2. The paper makes interesting observations relevant to the literature.


**Reasons To Reject:**

1. While the dataset proposed is a valuable resource to the community, the need for a semi-inductive link prediction setting is not well motivated. Also, it is not very well distinguished from existing transductive and inductive settings.


**Reproducibility:**

2: Would be hard pressed to reproduce the results. The contribution depends on data that are simply not available outside the author's institution or consortium; not enough details are provided.

**Reviewer Confidence:**

3: Pretty sure, but there's a chance I missed something. Although I have a good feel for this area in general, I did not carefully check the paper's details, e.g., the math, experimental design, or novelty.

---

> ### Author Rebuttal · Authors · 2023-08-28
>
> Thank you for your valuable comments and constructive criticism. We provide thoughts on each point below, and we will update the paper accordingly.
>
> > The paper does not provide enough motivation for why the SI-LP task is important.
>
> SI-LP focuses on modelling entities that are unknown or unseen during link prediction, such as out-of-KG entities (not part or not yet part of the KG) or newly created entities (e.g., a new user/product/event). A straightforward solution to handle unseen entities is to retrain the model and apply transductive inference. Unseen entities may arise frequently, however, and retraining the model is expensive for large KGs. The goal of SI-LP is to avoid retraining and perform LP directly, i.e., generalize beyond the entities seen during training. The goal of our benchmark is to provide a testbed for SI-LP methods. In addition, the benchmark clearly separates and evaluates different scenarios w.r.t. the information provided for unseen entities at test time: from zero- to few shot on the one hand and from no text to full textual descriptions on the other hand.
>
> > Does semi-inductive mean exactly one entity (either head or tail) is unseen during training in the zero-shot setting?
>
> Yes. The goal of semi-inductive LP is to reason how unseen entities connect to (the entities in) an existing KG.
>
> > Similarly, in few-shot, does exactly one entity is a low-frequency entity?
>
> No. In our benchmark, the unseen entity have low degree by construction. The other (seen) entitiy, can have low, mid, or high frequency. Some examples are shown in Tab. 5 (appendix) under "context triples", each of which is used as a prediction target in our evaluation.
>
> > Removing triples from a split can lead to a cascading effect, i.e., a reduction in other entities' frequencies. Was it one-shot removal or any measures were taken to handle the cascading effect?
>
> Since we focus on long-tail entities, the cascading effect is negligible and we did not take any particular countermeasure. Some detail: We removed 0.03% of entities and 0.07% triples from the train split. Moreover, long-tail entities are often connected to entities with a higher degree, and the resulting change of degree of these entities is relatively small.
>
> > Reproducibility: 2
>
> We will ensure reproducibility and publicly provide scripts, datasets, model implementations and hyperparameters of the presented baselines. (An excerpt of the benchmark dataset with a script for task generation is provided as additional material with this submission.)

---

### Official Review · Reviewer_WJjs · 2023-08-08

**Soundness:** 4

**Excitement:**

3: Ambivalent: It has merits (e.g., it reports state-of-the-art results, the idea is nice), but there are key weaknesses (e.g., it describes incremental work), and it can significantly benefit from another round of revision. However, I won't object to accepting it if my co-reviewers champion it.

**Missing References:**

This reference may be a good add to explain different transductive/inductive settings.

Ali et al. Improving Inductive Link Prediction Using Hyper-Relational Facts. In ISWC2021.

**Paper Topic And Main Contributions:**

This paper proposes a new large-scale benchmark for link prediction based on Wikidata5M. Specifically, the proposed Wikidata5M-SI focuses on semi-inductive and transductive link prediction, and ensures the unseen entities are long-tail entities. It allows to evaluate models with varying amount of context and provides controlled amount of textual information. It evaluates baselines with various settings and obtains some general conclusions of usefulness of each piece of information.

**Questions For The Authors:**

1. Why not also contain the inductive learning setting?

2. I am also wondering the performance of LLM in directly predicting the link with proper prompts.  I suppose LLMs will not be perfect but have a decent performance. If an LLM can already do this task well, the task will be not so meaningful and the authors have to think about how to make the dataset more useful.

**Reasons To Accept:**

1. Despite only a refinement of exsiting Wikidata5M, the proposed data did contribute to the large-scale KG reasoning community by providing another choice of dataset and settings.

2. The comparison of various settings is good for validating the usefulness of each type of information. It maybe inspires some future research.

**Reasons To Reject:**

1. The proposed benchmark is only a refinement of an existing data without much annotation effort. I admit it is useful for some specific tasks, but it is hard to evaluate its “novelty” as a dataset paper.

2. Some choices of building the dataset is not convincing enough. (1) why are the unseen entities long-tail? I believe if we just want to test the inductive learning ability of a model, it does not matter whether the unseen entity is long-tail or not.(2) entities with degree less than 10 are not considered. But it is quite practical to see such cases in real-world scenario.

3. The title is confusing. The dataset is not only for semi-inductive but also contains transductive.

4. I understant the space limit as a short paper. But if the authors have extra space, it is good to add a figure with examples to compare different settings.

**Reproducibility:**

3: Could reproduce the results with some difficulty. The settings of parameters are underspecified or subjectively determined; the training/evaluation data are not widely available.

**Reviewer Confidence:**

3: Pretty sure, but there's a chance I missed something. Although I have a good feel for this area in general, I did not carefully check the paper's details, e.g., the math, experimental design, or novelty.

---

> ### Author Rebuttal · Authors · 2023-08-28
>
> Thank you for your valuable comments and constructive criticism. We provide thoughts on each point below, and we will update the paper accordingly.
>
> > Why not also contain the inductive learning setting?
>
> The original Wikidata5M benchmark already contains a fully-inductive split, so there is no need. Moreover, the fully-inductive setting is very different to both transductive and semi-inductive LP; methods and applications differ considerably. In particular, fully-inductive LP evaluates the transfer of a trained model to a new KG that does not share *any* of the entities of the training KG (but only the relation types). Our focus, in contrast, is on providing a benchmark for new entities, i.e., entities that are unknown or unseen during link prediction. Examples include out-of-KG entities (not part or not yet part of the KG) or newly created entities (e.g., a new user/product/event).
>
> > I am also wondering the performance of LLM in directly predicting the link with proper prompts.
>
> The KGT5 and SimKGC models are SOTA approaches based on language modeling, and KGT5-context can be seen as a method to generate suitable prompts. We did not evaluate pretrained LLMs such as GPT-4 as these models are likely to have seen test data during training.
>
> > Why are the unseen entities long-tail?
>
> High-degree entities in a KG such as Wikidata often refer to types/taxons (e.g, human, organization, ...) as well as popular named entities (e.g., Albert Einstein, Germany, ...). These entities are fundamental to the KG and/or of high interest and have many facts associated with them. For this reason, they do not form suitable candidates for unseen or new entities. In addition, removing high-degree entities for the purpose of evaluating SI-LP is likely to distort the KG (e.g., consider removing type "human" or "Germany"). In contrast, our benchmark focuses on entities for which knowledge is not yet abundant: long-tail entities are accompanied by no or few facts (at least initially) and our SI-LP benchmark tests reasoning capabilities with this limited information.
>
> > Entities with degree less than 10 are not considered. But it is quite practical to see such cases in realworld scenario.
>
> We do study LP for entities with (known) degree less than 10 in our evaluation, in which we consider 0-shot up to 10-shot settings. The reason we selected entities with degree at least 10 in the original KG to form the test set is that we needed to have ground truth available. I.e., we can only evaluate 10-shot LP if 10 facts about the entity are actually known.
>
> > The title is confusing. The dataset is not only for semi-inductive but also contains transductive.
>
> The benchmark focuses on the semi-inductive setting. As you say, it can also be used to evaluate the transductive setting, but this setting can already be evaluated using the original Wikidata5M dataset. In fact, our benchmark aims to stay close to the original Wikidata5M transductive benchmark so that the performance of transductive evaluation is comparable across these two benchmarks.
>
> > It is good to add a figure with examples to compare different settings.
>
> Table 5 in the appendix shows an example of the varying amount of information (atomic, mention, description + a few facts). Is this what you had in mind? If not, additional suggestions are welcome!
>
> > Reference: Ali et al. Improving Inductive Link Prediction Using Hyper-Relational Facts. In ISWC2021.
>
> Thank you!

---

### Official Review · Reviewer_NEnw · 2023-08-12

**Soundness:** 3

**Excitement:**

3: Ambivalent: It has merits (e.g., it reports state-of-the-art results, the idea is nice), but there are key weaknesses (e.g., it describes incremental work), and it can significantly benefit from another round of revision. However, I won't object to accepting it if my co-reviewers champion it.

**Paper Topic And Main Contributions:**

The paper introduces a benchmark named Wikidata5M-SI for semi-inductive link prediction (LP) in knowledge graphs (KGs), specifically focusing on predicting facts for new, unseen entities. The benchmark provides a versatile evaluation framework for transductive, k-shot, and 0-shot LP tasks with varying levels of available information, including KG structure, textual mentions, and detailed entity descriptions. The study evaluates several recent approaches using this benchmark and finds that semi-inductive LP performance lags behind transductive LP across all experiments. The benchmark highlights the need for further research in integrating context and textual information for improving semi-inductive LP models. The authors compare different context selection strategies and explore the value of textual information in aiding LP predictions. They also identify trade-offs between TD and SI performance and discuss directions for future research.


**Reasons To Accept:**

- The paper introduces a new benchmark, Wikidata5M-SI, for semi-inductive link prediction in knowledge graphs. It addresses several important design goals, including focusing on long-tail entities, evaluating models with varying amounts of context and textual information, and providing a controlled setting for experimentation.
- The paper presents a thorough evaluation of various state-of-the-art models on the benchmark, providing insights into their performance in different scenarios.
- The study analyzes the impact of different types of contextual information (mentions, descriptions) and their integration into models for link prediction.
- The paper discusses trade-offs between transductive and semi-inductive performance and highlights the potential of leveraging textual information for improving link prediction in knowledge graphs.


**Reasons To Reject:**


- The paper lacks a clear motivation for why semi-inductive link prediction is an important problem and how it contributes to the broader field of knowledge graph research.
- The construction of the benchmark is simple and does not require any new techniques or methods. The paper does not provide a strong justification for why the benchmark is novel or important.
- The abstract is brief and does not provide a strong overview of the contributions, methodology, or results of the paper.
- The experimental study focuses on a single knowledge graph (Wikidata5M), which may limit the generalizability of the findings to other knowledge graphs.

**Reproducibility:**

5: Could easily reproduce the results.

**Reviewer Confidence:**

3: Pretty sure, but there's a chance I missed something. Although I have a good feel for this area in general, I did not carefully check the paper's details, e.g., the math, experimental design, or novelty.

---

> ### Author Rebuttal · Authors · 2023-08-28
>
> Thank you for your valuable comments and constructive criticism. We provide thoughts on each point below, and we will update the paper accordingly.
>
> > The paper lacks a clear motivation for why semi-inductive link prediction is an important problem and how it contributes to the broader field of knowledge graph research.
>
> Semi-inductve link prediction (SI-LP) focuses on modelling entities that are unknown or unseen during link prediction, such as out-of-KG entities (not part or not yet part of the KG) or newly created entities (e.g., a new user/product/event). A straightforward solution to handle unseen entities is to retrain the model and apply transductive inference. Unseen entities may arise frequently, however, and retraining the model is expensive for large KGs. The goal of SI-LP is to avoid retraining and perform LP directly, i.e., generalize beyond the entities seen during training. The goal of our benchmark is to provide a testbed for SI-LP methods. In addition, the benchmark clearly separates and evaluates different scenarios w.r.t. the information provided for unseen entities at test time: from zero- to few shot on the one hand and from no text to full textual descriptions on the other hand.
>
> > The construction of the benchmark is simple and does not require any new techniques or methods.
>
> We agree.
>
> > The paper does not provide a strong justification for why the benchmark is novel or important.
>
> Existing benchmarks lack one or more of the characteristics outlined in our paper: sensible selection of unseen entities, varying amount of contextual facts, varying amount of textual information, and large scale. The goal of our benchmark is to satisfy these critera and provide baseline results across the different settings; it is the first benchmark that does so. An additional goal was to stay close to the Wikidata5M transductive benchmark so that the performance of transductive evaluation is comparable.
>
> > The abstract is brief and does not provide a strong overview of the contributions, methodology, or results of the paper.
>
> We will expand the abstract accordingly.
>
> > The experimental study focuses on a single knowledge graph (Wikidata5M), which may limit the generalizability of the findings to other knowledge graphs.
>
> We agree; this is a limitation of our work. We will provide scripts that allow for creation of similar benchmarks for other knowlege graphs, which facilitates the creation of benchmarks for other KGs in future research.

---

### Meta-Review · Area_Chair_3AqH · 2023-09-19

**Recommendation:** 3

**Metareview:**

This paper introduced a new large-scale benchmark for link prediction derived from Wikidata5M. This dataset focuses on fact prediction for unseen entities given a variety of contextual information.

Reviewers appreciated the importance of the problem space, and detailed evaluation presented in this paper, but they concerned about the motivation of semi-inductive link prediction and dataset construction.

---

### Decision · Program_Chairs · 2023-10-07

**Decision:**

Accept-Findings

**Comment:**

This paper introduced a new large-scale benchmark for link prediction derived from Wikidata5M. This dataset focuses on fact prediction for unseen entities given a variety of contextual information.

Reviewers appreciated the importance of the problem space, and detailed evaluation presented in this paper, but they concerned about the motivation of semi-inductive link prediction and dataset construction.